# Engineering Extracellular Microenvironment for Tissue Regeneration

**DOI:** 10.3390/bioengineering9050202

**Published:** 2022-05-08

**Authors:** Dake Hao, Juan-Maria Lopez, Jianing Chen, Alexandra Maria Iavorovschi, Nora Marlene Lelivelt, Aijun Wang

**Affiliations:** 1Department of Surgery, School of Medicine, University of California Davis, Sacramento, CA 95817, USA; dkhao@ucdavis.edu (D.H.); jmglopez@ucdavis.edu (J.-M.L.); jnichen@ucdavis.edu (J.C.); amiavorovschi@ucdavis.edu (A.M.I.); nmlelivelt@ucdavis.edu (N.M.L.); 2Institute for Pediatric Regenerative Medicine, Shriners Hospitals for Children, Sacramento, CA 95817, USA; 3Department of Biomedical Engineering, University of California Davis, Davis, CA 95616, USA

**Keywords:** extracellular matrix, extracellular vesicles, growth factors, cell behaviors, tissue regeneration

## Abstract

The extracellular microenvironment is a highly dynamic network of biophysical and biochemical elements, which surrounds cells and transmits molecular signals. Extracellular microenvironment controls are of crucial importance for the ability to direct cell behavior and tissue regeneration. In this review, we focus on the different components of the extracellular microenvironment, such as extracellular matrix (ECM), extracellular vesicles (EVs) and growth factors (GFs), and introduce engineering approaches for these components, which can be used to achieve a higher degree of control over cellular activities and behaviors for tissue regeneration. Furthermore, we review the technologies established to engineer native-mimicking artificial components of the extracellular microenvironment for improved regenerative applications. This review presents a thorough analysis of the current research in extracellular microenvironment engineering and monitoring, which will facilitate the development of innovative tissue engineering strategies by utilizing different components of the extracellular microenvironment for regenerative medicine in the future.

## 1. Introduction

Tissue regeneration combines the concepts from material science, bioengineering principles, and transplantations, with engineering designs to restore, maintain, and improve cell functions for the regeneration of injured tissues [1,2]. Currently, different tissue engineering approaches have been established, such as cell engineering and the biomaterial-based cell delivery system [3,4,5]. Regardless of the treatment approach for injured tissues, the presence of cellular components is inevitable. Native tissue accumulates reservoirs of cells, which remain quiescent and retain their plasticity until they are needed in the body for their therapeutic properties. The extracellular microenvironment controls the cells’ plasticity and quiescence and their subsequent functions, such as survival, proliferation and section, which indicates that mimicry of the extracellular microenvironment marks the most technically viable supply source [6,7]. However, the extracellular microenvironment is a dynamic mixture of biophysical and biochemical cues, which poses challenges for current tissue engineering designs [8]. Therefore, to develop the engineering approaches necessary to improve cell functions for tissue regeneration, it is essential to understand the fundamental components of the extracellular microenvironment.

Cells reside in a complex extracellular microenvironment, also defined as cell niche, composed of the extracellular matrix (ECM), extracellular vesicles (EVs), and growth factors (GFs), all of which play different key roles in determining the biological processes of cells (Figure 1) [9,10,11]. Thus, dissecting the composition of the extracellular microenvironment and understanding the diverse effects that each component has on cell activities is crucial when designing engineering strategies to mimic the dynamic extracellular microenvironment. This deeper understanding can then provide insight into promising solutions for the next generation of tissue engineering and translate into long-term success for future clinical products [12]. In this review, we discuss the correlative actions of different components of the extracellular microenvironment, as well as the approaches to engineer native-mimicking components for tissue regeneration.

## 2. Extracellular Matrix (ECM)

### 2.1. Native ECM

The ECM is composed of a complex three-dimensional (3D) interconnecting network of a variety of proteins, such as collagen, fibronectin, laminin, elastin, and proteoglycans (Figure 2) [13,14]. Collagen is the main structural protein in natural ECM, which can mimic the ECM physical characteristics [15]. Fibronectin is a high-molecular-weight (~500–~600 kDa) ECM glycoprotein that binds to membrane-spanning receptor proteins called integrins, which could opsonize the migration, proliferation, and contraction of cells during the healing process. Fibronectin also binds to other ECM proteins, such as collagen, fibrin, and heparan sulfate proteoglycans (e.g., syndecans) [16,17,18]. Laminins are high-molecular-weight (~400 to ~900 kDa) proteins of the extracellular matrix. They are a major component of the basal lamina (one of the layers of the basement membrane), a protein network foundation for most cells and organs. The laminins are an important and biologically active part of the basal lamina, influencing cell differentiation, migration, and adhesion [19,20]. Proteoglycans are present during the development of the central nervous system and contain membrane bound proteins that interact with different cellular microenvironment molecules [21]. Lecticans, a type of chondroitin sulfate proteoglycans (CSPG), is the most abundant CSPG secreted during the development of the central nervous system [22,23]. Heparan sulfate proteoglycans are an important component of the glomerular basement membrane that can interact with growth factors and secret proteins, containing syndecan and glypican, to mediate these interactions [24]. This network of proteins associates with each other to regulate different cellular processes, including differentiation, growth, and survival [25,26,27]. The combination of ECM, cells, and receptors creates microenvironmental signaling pathways that will impact cell fate [28,29,30]. 

Native ECM has a highly dynamic structure and constantly undergoes a remodeling process where deposition, degradation, and modification occur [31,32]. These changes are fundamental in the regulation of cell differentiation, angiogenesis, wound healing, and in the formation of stem cell niches [32,33]. Conversely, deregulation of the ECM dynamic environment can be potentially harmful, as it can cause abnormal behavior of the stem cell abilities to differentiate and proliferate consequently, leading to a potentially tumorigenic microenvironment and pathological diseases [34,35]. Therefore, careful attention to understanding ECM dynamics and control is critical in developing more efficient therapeutic interventions for related cellular and tissue disorders.

### 2.2. Engineering ECM for Tissue Regeneration

ECM components have been widely applied in tissue engineering approaches, based on the ability of the ECM to regulate and improve cell functions during tissue regeneration [36]. ECM-derived hydrogel systems and porous scaffolds have been applied in the field of stem cell transplantation, due to their 3D cross-linked networks, cytocompatibility, injectability, and biocompatibility (Figure 2) [37,38,39,40,41]. Specifically, the ubiquitous type I collagen has considerable potential as a cell delivery medium because of its self-assembly capability under physiological conditions [42,43,44]. Fibronectin matrix has been developed as a scaffold for procollagen proteinase binding and collagen processing [45]. Laminins have been applied to functionalize different types of 3D biomimetic scaffolds for the regeneration of tissues, such as nerve [46,47], skeletal muscle [48,49], and blood vessels [50]. It was shown that decorin, a small leucine-rich proteoglycan, was involved in regulating the prevention of hypertrophic scars and collagen fibrillogenesis [51]. A proteoglycan mimetic composed of conjugated dermatan sulfate (DS) backbone and collagen binding peptide, SILY, was engineered to bind to collagen matrices protecting the matrix from rapid proteolytic decomposition in wounds [52]. Collagen, the main structural ECM protein, is generally applied to develop the structure of the scaffold, whereas fibronectin, laminin, and proteoglycans mainly contribute to biologically functionalizing the scaffolds [53]. Although collagen possesses the ECM-mimicking structure, its specific biological functions in tissue regenerative applications still require further optimization. Collagen-based tissue engineering strategies are currently being studied in tissue engineering [53]. Many different biological molecules have been developed to functionalize the collagen-based scaffolds to improve their regenerative potentials. Integrin-ligand binding facilitates cell-ECM adhesion, activates signal transduction pathways, and regulates cell functions. Our recent study has focused on increasing the specific integrin binding sites of the collagen-based scaffolds to advance the endothelial cell (EC)/endothelial progenitor cell (EPC) functions and improve vascularization. We identified a ligand LXW7 that specifically binds to integrin αvβ3 and found that the LXW7-modified collagen hydrogel significantly improved EC spreading, proliferation, and survival in vitro and the engraftment of transplanted ECs in vivo [54]. Our group also has immobilized the LXW7 onto the collagen-based small intestinal submucosa (SIS) scaffolds and demonstrated that the LXW7 modified SIS scaffold increased EPC retention and enhanced angiogenic functions, leading to improved healing at the ischemic site [55]. In other studies, Malcor et al. have established an approach to modify the collagen scaffolds with triple-helical peptides to support 3D EC culture [56]. Amaral et al. have demonstrated that collagen-based scaffolds functionalized with platelet-rich plasma could enhance skin wound healing [57]. Li et al. have successfully achieved the use of collagen scaffolds functionalized with neutralizing proteins to facilitate spinal cord regeneration [58]. Thus, the components of ECM could be endowed superior regenerative potentials via various engineering strategies.

### 2.3. Engineering Artificial ECM-Mimicking Scaffolds for Tissue Regeneration

One goal for advancing the field of tissue engineering is to develop an artificial network, by mimicking the functional aspects of native ECM, including cell adhesion, control over cell behaviors, and proteolytic degradation processes [59]. Advancements in artificial ECM network development for modeling mutations or changes in protein regulation and related developmental illnesses are known as the protein expression profile [60]. Artificial ECM networks tend to become responsive towards cell-mediated signaling and proteolytic remodeling, a process involving enzymatic interactions that enable the cells to alter their relationship with their microenvironment by cleaving off structural molecules of the ECM [61]. Mediation of interactions with other molecules involves both the central protein and the carbohydrate structures. The ECM consists of various topographical structures, including nanocrystals, nanofibers, and nanopores, influencing biochemical signaling, and cell functions [62]. Various strategies, such as electrospinning and 3D printing, have been used to engineer artificial ECM structures mimicking scaffolds to fulfill tissue regeneration related advancements (Figure 2) [63,64]. In addition, cell signaling occurs when adhesion molecules, such as integrin ligands, and cell surface growth factor receptors are present in combination with the ECM to create optimum environment [65]. Thus, various functional molecules have been developed to functionalize the artificial ECM structure and mimic scaffolds to further improve their regenerative potentials. Recently, we have identified a ligand LLP2A that has the high binding-affinity for integrin α4β1 and used it to modify the electrospun polyester scaffolds. It was demonstrated that the LLP2A modification improved the adhesion and proliferation of mesenchymal stem cells (MSCs) on the electrospun polyester scaffolds and also activated the integrin-mediated signals, such as focal adhesion kinase (FAK) [66]. We have also identified another ligand, LXW7, which specifically targets integrin αvβ3. LXW7 was used to construct the electrospun vascular grafts and it was demonstrated that the LXW7 modified electrospun vascular grafts improve the adhesion and proliferation of EPCs/ECs and improved rapid endothelialization and long-term patency in a rat carotid artery model [67,68]. Zheng et al. have established an approach to modify the electrospun vascular grafts with RGD peptide and showed an improved remodeling and integration capability in revascularization in a rabbit carotid artery model [69]. Yu et al. have successfully achieved the use of electrospun vascular grafts, functionalized with stromal cell-derived factor-1α and heparin to accelerate vascular healing [70]. Yang et al. have successfully constructed the osteoinductive 3D-printed scaffold with bone morphogenetic protein 2 (BMP-2) and healed 5 cm segmental bone defects in the ovine metatarsus [71]. Wang et al. have successfully constructed the cryogenic 3D printing of dual-delivery scaffolds with BMP-2 and vascular endothelial growth factor (VEGF) for improving bone regeneration with enhanced vascularization [72]. It was known that matrix-bound vesicles (MBVs) were identified as an integral and functional component of ECM [73,74]. To further explore the functionalization of the scaffolds, we recently developed an approach to immobilize the MSCs-derived extracellular vesicles (EVs, MSC-EVs) onto the electrospun scaffolds to improve the angiogenic potentials of the scaffolds [75]. An ideal artificial ECM could not only mimic the structure of the native ECM, but also possesses the biological functions of native ECM. 

### 2.4. Advantages and Disadvantages of Current Engineering Approaches in ECM

The current engineered ECM scaffolds have advantages in providing the similar features, mechanical properties and biological components of their original tissue or organ, which facilitate their applications in the specific diseases. However, the native ECM scaffolds have limited autologous tissue or organ sources, and the allogeneic or xenogeneic native ECM scaffolds have the risk of host immune responses [76]. The engineered artificial ECM-mimicking scaffolds overcome the limited tissue or organ sources and allow the controllable shapes for targeted tissue regeneration. In addition, different engineering approaches have been designed to enable the artificial ECM-mimicking scaffolds with the similar mechanical and biological properties [77]. However, the native ECM is a complex dynamic environment, much research remains to be done on the interactions between the components of the native ECM to provide more support for improving the regenerative capacities of the artificial ECM-mimicking scaffolds.

## 3. Extracellular Vesicles (EVs)

### 3.1. Native EVs

EVs are lipid bilayer-delimited vesicles that are naturally released from cells. EVs range in diameter from around 20–30 nm to as large as 10 μm, although most EVs are smaller than 200 nm. EVs can be divided according to size and synthesis route into exosomes, microvesicles, and apoptotic bodies (Figure 3) [78]. Exosomes range in size from 30–200 nm in diameter. Biogenesis is the process by which EVs are release from cells and the process begins with pinching off endosomal invaginations into the multivesicular body (MVB) [79]. MVBs act as trafficking vesicles for cell materials, where cargo, including early endosomes, is sent to and released at the plasma membrane until eventually secreted into the extracellular space [80]. Exosomes are involved in cell–cell and cell–ECM communication and could also trigger an immune response by presenting antigens within them [81]. Thus, exosomes are being used as therapy delivery tools, therapeutic targets, and biomarkers [82]. Exosomes have been shown as GTP-activatable phospholipases and lipid mediators in cell-to-cell interactions. In vaccine research, exosomes are being studied for use as a powerful immune regulator when engineered with specific mRNAs [83]. For use as biomarkers, exosomes are being studied in cancer biology because exosomes are involved in the development of cancer through the horizontal transfer of genetic information between cancer cells [84]. Microvesicles can be as small as the smallest EVs (30 nm in diameter) or as large as 1000 nm, and play a key role in intercellular communication and transportation of molecules such as mRNA, miRNA, and proteins between cells [85]. Microvesicles also have been implicated in numerous physiologic processes, including anti-tumor effects, tumor immune suppression, metastasis, tumor-stroma interactions, angiogenesis, and tissue regeneration [86,87,88]. Apoptotic bodies were previously regarded as garbage bags, and recently they have also been found to possess the capacity for delivering cargoes to healthy recipient cells [89]. Based on their structure and mode of function, these three types of EVs have potential in tissue engineering approaches.

### 3.2. Engineering Native EVs for Tissue Regeneration

EVs derived from various types of cells have shown therapeutic potentials for different kinds of diseases, such as ischemia injury [90,91], wound healing [92,93], graft-versus-host disease [94], Alzheimer’s disease [95], arthritis [96], bone defects [97,98], liver disease [99], kidney injury [100], pancreatic islet transplantation [101], and pulmonary hypertension [102]. However, the therapeutic potentials of EVs are mainly limited by their short half-life of approximately few minutes and low local retention after transplantation [103,104,105]. Currently, various engineering strategies have been designed to improve the therapeutic capacity of EVs (Figure 3). Some studies have focused on engineering EVs with specific targeting molecules to improve the distribution of transplanted EVs in the specific disease area and augment their therapeutic efficiency [106]. To improve the efficiency with which EVs home precisely onto their target cells, one approach is to engineer the EV surfaces with targeting peptides by overexpressing the peptides in the EV-donor cells via transfection. Alvarez-Erviti et al. have successfully generated brain-targeting EVs from dendritic cells, transfected with a plasmid encoding EV protein Lamb2b and brain-targeting RVG peptide, and they showed that the engineered EVs could cross the blood–brain barrier (BBB) and deliver RNA into the brain [107]. Although this genetic engineering approach can enable stable conjugation of EVs with the peptides, it is timely, cost consuming and poses a high risk of horizontal gene transfer. An alternative approach is to functionalize the EV surface with targeting peptides post-isolation via chemical or affinity-based methods. Tian et al. functionalized EV surfaces with Arg-Gly-Asp (RGD) peptide to improve EV targeting and therapeutic efficiency in cerebral ischemia therapy [108]. Cui et al. have demonstrated that rabies viral glycoprotein (RVG)-modified MSC-EVs rescue memory deficits by regulating inflammatory responses in Alzheimer’s disease [109]. Another strategy for improving the targeting efficacy of EVs is the fusion of EVs with liposomes or lipid-based micelles, which means the lipid components of the EV membrane fuse seamlessly with the synthetic lipid vesicles to obtain exogenous functional lipids or peptides. Sato et al. have successfully fused the EVs isolated from Raw264.7 and CMS7 cancer cells with liposomes by using the freeze-thaw method [110]. Evers et al. have constructed the hybrid cardiac progenitor cell (CPC) derived-EVs by fusing with liposomes, and they showed these hybrid EVs improved uptake efficiency and delivery, wound healing, and phosphorylation of Akt that play key roles in multiple cellular processes, such as cell apoptosis, cell proliferation, and cell migration, compared to liposomes alone [111]. Additionally, some studies established biomaterial-based EV delivery systems for promoting EV local retention and stability after transplantation to augment the therapeutic efficiency of EVs [112]. Li et al. have immobilized the EVs derived from human adipose-derived stem cells (hASCs) on the polydopamine-coating poly(lactic-co-glycolic acid) (PLGA/pDA) scaffolds by using mild chemical conditions to augment the bone regenerative capacity of EVs [113]. Zhang et al. have incorporated human placenta-derived MSC-derived EVs with chitosan hydrogel to enhance the retention and stability of EVs and further enhance their therapeutic effects in hindlimb ischemia treatment [114]. Thus, improving EV targeting via various approaches has the potential to improve tissue regeneration by improving the targeting efficiency, retention, and stability of EVs at the injury sites. 

Cargo engineering is another approach to regulate and enhance EV functions in therapeutic applications. Two main strategies have been designed to engineer the cargoes in EVs. One strategy is preconditioning or genetically engineering the parent cells of the EVs [115,116,117]. Xu et al. have shown that EVs generated from bone marrow MSCs treated with lipopolysaccharides have a greater efficacy in attenuating inflammation and driving macrophage polarization to a more anti-inflammatory M2 phenotype, and they markedly reduced post-infarction inflammation and improved cardiomyocyte survival and recovery in a murine acute myocardial infarction model [118]. Wu et al. have demonstrated that EVs produced from human cardiac progenitor cells (CPCs) with hypoxic preconditioning included more lncRNA MALAT1 and enhanced EC viability, reduced cardiomyocyte apoptosis, and improved vascularization, compared to normoxic CPC-derived EVs [119]. Gong et al. have produced EVs with high levels of stromal cell-derived factor 1α (SDF-1α) from MSCs transfected with SDF-1α plasmids and showed these EVs inhibited ischemic myocardial cell apoptosis and promoted cardiac endothelial microvascular regeneration in mice with myocardial infarction [120]. The other cargo engineering strategy consists of loading the exogenous cargoes into EVs via mechanical or chemical techniques, such as saponin treatment, sonication, and electroporation to improve their therapeutic functions. Fuhrmann et al. have demonstrated saponin could increase hydrophilic drug loading into EVs by 11-fold compared to passive incubation [121]. Haney et al. showed sonication could improve TPP1 loading efficiency in EVs derived from IC21 macrophages by roughly 30% and enhance brain distribution in the batten disease model compared to saponin permeabilization [122]. Ma et al. have demonstrated that microRNA-132 could be loaded into MSC-derived EVs via electroporation, and the EVs loaded with microRNA-132 promoted angiogenesis in myocardial infarction [123]. Surface modification strategies could also be applied in conjunction with cargo engineering strategies to achieve a targeted delivery of specific molecules.

### 3.3. Engineering Artificial Evs for Tissue Regeneration

Although EVs have been shown to have exceptional therapeutic potentials in tissue regeneration, there is still a limitation due to the finite knowledge of their signaling pathways and complex heterogenic of molecular and physical structure. Additionally, many challenges, including, mass-production, complex isolation and purification, stability, and quality control of native EVs have been limiting to EV clinical translation [124]. Therefore, different types of artificial EVs, such as nanovesicles and EV mimics, have been designed by top-down and bottom-up strategies, which are powerful alternatives to natural EVs for tissue regeneration (Figure 3). The top-down strategy is designed to disassemble big parent cells to form smaller nanovesicles. Because nanovesicles are derived from patent cells, they possess natural nucleic acids, proteins and lipids, which mimic the biological complexity of native EVs with low immunogenicity [125]. The top-down strategy can be achieved via different methods, such as the extrusion-based method, filtration-based method, microfluidic device-based method, nitrogen cavitation-based method, sonication-based method, and cell bleb-based method. Kim et al. developed nanovesicles by extruding adipose stem cells (ASCs) through filters with gradient apertures and showed the nanovesicles possess similar beneficial effects in animals with emphysema, compared to the ASCs [126]. Kim et al. also developed MSCs-derived nanovesicles that achieved great therapeutic potential in spinal cord injury treatment [127]. Gao et al. generated neutrophil-derived nanovesicles by using the nitrogen cavitation method and demonstrated that the nanovesicles could mitigate acute lung inflammation [128]. However, nanovesicles may have less heterogeneity, and their construction is also a time-consuming process [129,130]. Thus, fully synthetic EV-mimics are being constructed using the bottom-up strategy of functional and controlled lipids, proteins, and RNA to overcome the limitations of native EVs and nanovesicles. The bottom-up strategy is a manufacturing approach that begins with small molecules forming large and complex structures through a stepwise assembling process. The bottom-up strategy possesses high-encapsulation efficiency of the membrane, integral proteins, and biomolecules, as well as size homogeneity in the construction of EV mimics. Moreover, the biophysical and biochemical composition of EV mimics can also be precisely quantitatively controlled, allowing researchers to create EV mimics that are uniformly and compositionally pure, thus providing a metric for future dosing of clinical products and studying the biophysical mechanisms of native EVs. Staufer et al. developed a bottom-up EV mimic with all the molecular and proteomic composition of EVs from human fibrocytes and showed the EV mimic significantly augmented wound healing, compared to the negative control and showed no significant difference in the positive control of native EV treated wounds [131]. Martinez-Lostao et al. generated liposomes conjugated with APO2L/TRAIL and demonstrated these EV mimics showed considerable therapeutic effects on arthritis in rabbits [132]. Vazquez-Rios et al. developed a liposome-based EV mimic that simulates structure and functions of native EV by loading therapeutic oligonucleotides and tailoring with integrin α6β4 for targeted drug delivery to lung adenocarcinoma cells [133]. Engineering artificial EVs have been shown to have exceptional therapeutic potentials in tissue regeneration; therefore, different types of artificial EV mimic creation, have been designed by top-down and bottom-up strategies as powerful alternatives to natural EVs for tissue regeneration.

### 3.4. Advantages and Disadvantages of Current Engineering Approaches in EVs

The engineering approaches for pre-isolation modification of natural EVs have the potential for improving the targeting, tracking, or pharmaceutical activities of EVs, but they are confined to the proteins and peptides expressed on the EV surface [134]. The post-isolation surface engineering approaches can be applied to immobilize any type of molecules or ligands onto the EV surface and also maintain the most biophysical properties of EVs; however, the crucial challenge is the removal of unincorporated materials [135]. Therefore, the suitable purification methods need to be developed, which will not only ensure the purity of the modified EVs, but also maintain the integrity and activity of the modified EVs. In addition, among the cargo loading techniques, transfection is advantageous over electroporation, with a higher loading efficiency and molecular stability, but is risked by toxicity and safety concerns which might also cause changes in the EV cargo and bioactivity by the transfection reagents [136]. When engineering artificial EVs through the top-down approaches, the microfluidic system has an advantage in its simplicity and advanced fabrication, especially in detecting, purifying, and engineering nanosized materials. However, the top-down approaches exhibit a limit in heterogeneity and the purification process is time-consuming [124]. Contrarily, the bottom-up engineering approaches are rather straightforward and allow for adjustment of individual EV compositions and creation of rare EV compositions [131]. 

## 4. Growth Factors (GFs)

### 4.1. Native GFs

GFs are secreted from cells and interact directly with the ECM (Figure 4). Coordinating the interactions between GFs, cells, and the ECM is crucial to define a localized cellular microenvironment that optimizes cell and tissue growth [137]. GFs typically act as signaling molecules between cells. Specific GFs are particularly capable of inducing and improving cell growth, proliferation, renewal, adhesion, migration, and differentiation [138]. For example, epidermal growth factor (EGF) enhances osteogenic differentiation, while fibroblast growth factor (FGF) and vascular endothelial growth factor (VEGF) stimulate blood vessel differentiation [139]. Although growth factors show promising advantages and applications, they exhibit limitations from their short half-life. They can also be enzymatically deactivated and have low retention in the extracellular microenvironment [140]. 

### 4.2. Engineering GFs for Tissue Regeneration

To maximize the function and efficiency of GFs for use in tissue regeneration, developing methods of control over their concentration and retention is crucial. One approach to improve the local retention of GFs after transplantation is endowing the molecule with binding potential by conjugating the molecule to target a specific cell or tissue, which could assist the GFs to remain in the targeted area and improve the local concentration of GFs (Figure 4). Zhang et al. have developed a collagen-targeting VEGF and showed it improved cardiac performance after myocardial infarction [141]. Sun et al. constructed a collagen-binding SDF-1α and showed it enhanced cardiac function after myocardial infarction by recruiting endogenous stem cells [142]. Another approach to improve the stability and local concentration of GFs is by constructing a GF control/release system using biomaterials (Figure 4). Immobilizing GFs on biomaterials through either a covalent or noncovalent approach is one way to achieve control of these GFs [138]. The release of the GFs is realized by breaking the bonds, utilizing hydrolysis, or enzymatic reactions. Chiu et al. have immobilized VEGF and angiopoietin-1 (Ang1) on porous collagen scaffolds by using 1-ethyl-3-(3-dimethylaminopropyl)carbodiimide hydrochloride (EDC) chemistry. They showed that the immobilized GFs resulted in higher cell proliferation and lactate metabolism than soluble GFs used at comparable concentrations [143]. Ikegami et al. immobilized bFGF on the heparin-conjugated collagen scaffold and showed the immobilized bFGF had twice higher stability than the bFGF solution [144]. Moreover, the direct interaction between ECM and GFs offers protection from degradation of the GFs [145]. ECM regulates the retention and presentation of GFs through electrostatic interactions. Electronegative components within the ECM, such as glycosaminoglycans etc.), possess a high binding affinity for the amine groups with positive charge on GFs [146]. Therefore, strategies have been designed to generate biomaterials with increased electronegative charge to mimic the high affinity of GFs to the ECM. Facca et al. demonstrated the presentation of bone morphogenetic protein 2 (BMP-2). Transforming growth factor beta 1 (TGF-β1) from polyelectrolyte (PE)-coated scaffold with a high negative electrostatic charge induced osteogenic differentiation of embryonic stem cells after subcutaneous implantation in mouse [147]. Shah et al. showed that BMP-2- and VEGF-loaded polyelectrolyte multilayer (PEM) films promoted bone regeneration upon subcutaneous implantation in rat [148].

### 4.3. Engineering Artificial GF Mimics for Tissue Regeneration

The engineering of artificial GFs is focused on mimetic peptides, which are made to mimic bioactive regions of native GF structures and can then mimic function (Figure 4). GFs can degrade when transplanted in vivo due to proteolytic activity. Rapid diffusion away from the target delivery source will lead to the administration of large concentrations [149]. Mimetic peptides can be more advantageous in regenerative medicine because they are more stable than the whole GF, due to their linear structure that allows for covalently boned linkers modifications without losing activity. Mimetic peptides can be engineered to be localized in a target location for a modulator release. For example, it was shown that transforming growth factor-beta (TGF-beta) regulates embryogenesis, growth, differentiation, and wound healing on a cellular level [150]. Bioactive peptides were developed to mimic the effects of TGF-beta, cytomudulin-1, and cytomodulin-2. These were synthesized to simulate the binding domain of TGF-beta and then validated in vitro on human foreskin fibroblasts. The results of this study showed that both CM-1 and CM-2 accelerated affected wound healing and cell migration, compared to the control of no peptide. It also showed the induction of collagen I production, demonstrating CM-1 and CM-2 biological activity similar to TGF-beta. FGF-2 is a cytokine that induces healing processes in cartilage and bone regeneration. Heparin-binding regions of FGF-2 have been shown as a potential for osteogenic regeneration; specifically, F105 and F129 binding domains peptides were shown to be stable and immobilized onto tissue culture plates and were shown to have increased binding affinity to heparin compared to a control of nonbinding peptides [151]. These mimetic peptides can be utilized as tools for surface modification of tissue engineering scaffolds to promote tissue regeneration.

### 4.4. Advantages and Disadvantages of Current Engineering Approaches in GFs

The approaches for engineering the natural GFs, such as conjugating the targeting molecules and establishing the control release system, could enhance the regenerative efficiency of the natural GFs by improving their retention at the target area, and could also retain the bioactivities of the natural GFs during incorporation [152]. However, the natural GFs still exhibit immunogenicity and limited stability in vivo. GF mimics possess the advantages in terms of longer half-life, lower immunogenicity, and crossing the blood–brain barrier, and they also can be administered orally [153]. However, different peptides have different functional limitations, so it is important to find the appropriate molecules or peptides to use in order to maximize the functionality.

## 5. Conclusions

The extracellular microenvironment is defined as the dynamical environment surrounding cells. Controlling the extracellular microenvironment remains an important goal to direct cell behavior and promote tissue regeneration. Currently, various strategies have been designed to optimize the native components of extracellular microenvironment and construct the artificial components of microenvironment via mimicking the structure of the native components and interactions between the cells and microenvironment. During the process of engineering, the extracellular microenvironment and new tissue engineering products came into being [154], including a tissue-engineered tracheal, elastic cartilage, etc., providing a great contribution to tissue regeneration and clinical applications. Further exploration of the biological process that occurs in the natural extracellular microenvironment is promising for establishing innovative and effective engineering approaches to promote tissue regeneration. The bottom-up design of modular biomimetic products holds the promise to produce multifunctional and multicellular structures, real time biosensing that could enable the analysis to be carried out at point-of-care (POC), as well as bioactive agent delivery systems to recapitulate the intricate extracellular microenvironment and improve the construction of future artificial tissues.

## Figures and Tables

**Figure 1 bioengineering-09-00202-f001:**
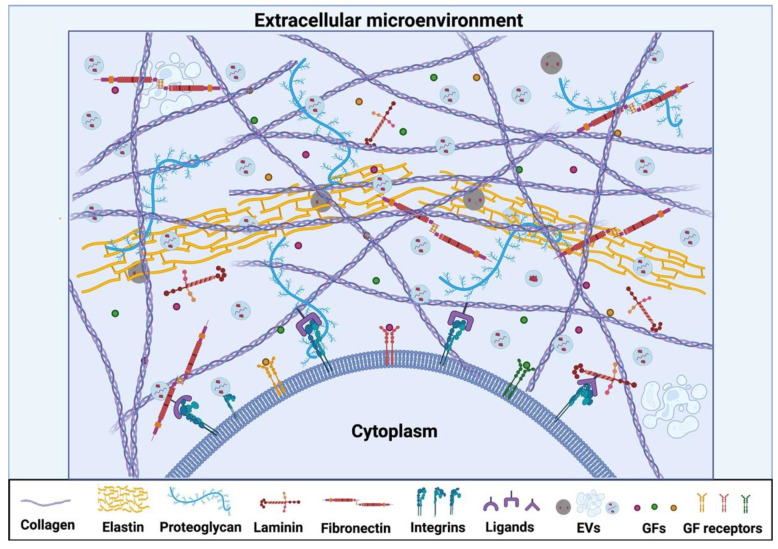
Schematic diagram of extracellular microenvironment.

**Figure 2 bioengineering-09-00202-f002:**
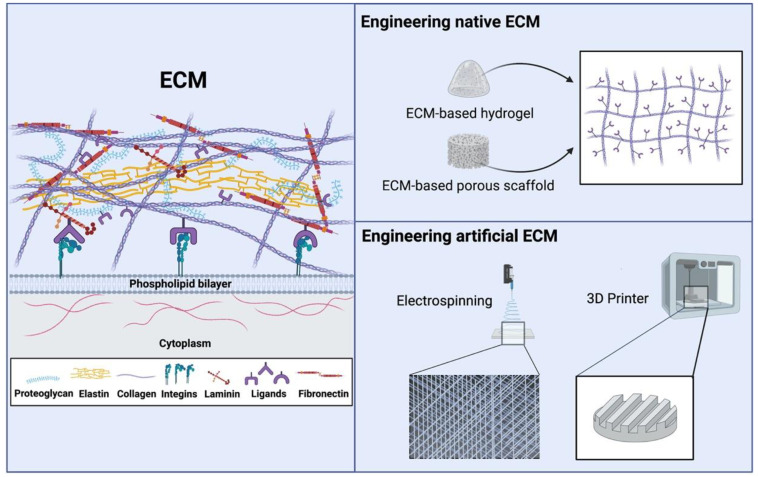
Engineering native ECM and artificial ECM.

**Figure 3 bioengineering-09-00202-f003:**
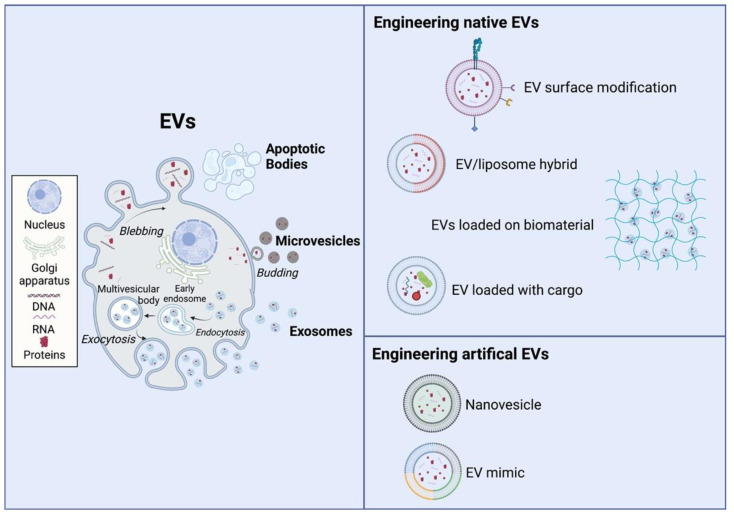
Engineering native EVs and artificial EVs.

**Figure 4 bioengineering-09-00202-f004:**
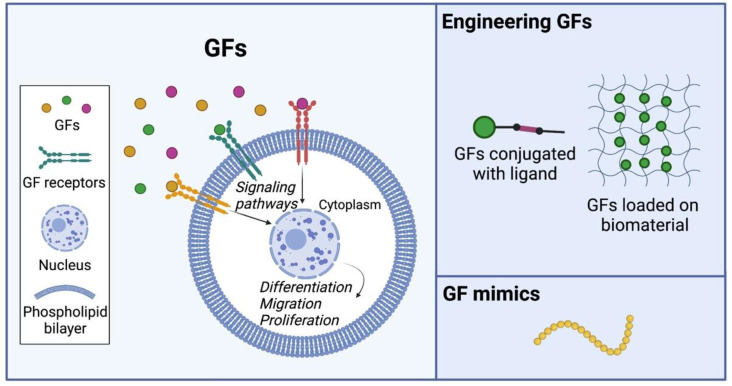
Engineering GFs and artificial GF mimics.

## Data Availability

Not applicable.

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
