# Peer review of "Engineering Extracellular Microenvironment for Tissue Regeneration"

_bioengineering, 2022, doi:10.3390/bioengineering9050202_

Round 1

Reviewer 1 Report

The review manuscript bioengineering-1674539 titled “Engineering Extracellular Microenvironment for Tissue Regeneration” submitted by A. Wang and co-authors means to give an overview of the factors influencing the extracellular microenvironment, providing it with a highly dynamic network of biophysical and biochemical cues able to drive cell behaviour, organisation, evolution and, ultimately, functionality. 

The Review focuses on just three key components of the extracellular microenvironment, that are the extracellular matrix (ECM), extracellular vesicles (EVs) and growth factors (GFs),  highlighting some of the more recent engineering approaches to exploit these components for advanced tissue regeneration purposes. 

Authors did a great job, resulting in a very well organised and concise review. The topic of the review is covering a critical aspect of bioengineering and tissue regeneration, consequently this work could represent a valuable reference for the research in these fields. I do recommend its publication in the Bioengineering journal.

Author Response

Reviewer 1:

The review manuscript bioengineering-1674539 titled “Engineering Extracellular Microenvironment for Tissue Regeneration” submitted by A. Wang and co-authors means to give an overview of the factors influencing the extracellular microenvironment, providing it with a highly dynamic network of biophysical and biochemical cues able to drive cell behavior, organization, evolution and, ultimately, functionality.

The Review focuses on just three key components of the extracellular microenvironment, that are the extracellular matrix (ECM), extracellular vesicles (EVs) and growth factors (GFs), highlighting some of the more recent engineering approaches to exploit these components for advanced tissue regeneration purposes.

Authors did a great job, resulting in a very well organized and concise review. The topic of the review is covering a critical aspect of bioengineering and tissue regeneration, consequently this work could represent a valuable reference for the research in these fields. I do recommend its publication in the Bioengineering journal.

Response: We really appreciate the reviewer's assessment and recognition of this work.

Reviewer 2 Report

The manuscript by Hao et al. reviewed extracellular microenvironments, such as ECM, EV, and GFs, in the native and engineered forms to provide strategies in control of cellular activities and tissue regeneration. Overall, the review manuscript was clear and well written. Please consider addressing the following comments to further enhance the quality of the manuscript.

  1. Please enhance the quality of Figure 1 - 4. Please provide references for elements not generated by the authors.
  2. It is recommended that the authors to provide a paragraph of comments/reviews on the advantages/disadvantages of current engineering approach in ECM, EV, and GFs to control cellular activities and tissue regeneration.

Author Response

Reviewer 2:

The manuscript by Hao et al. reviewed extracellular microenvironments, such as ECM, EV, and GFs, in the native and engineered forms to provide strategies in control of cellular activities and tissue regeneration. Overall, the review manuscript was clear and well written. Please consider addressing the following comments to further enhance the quality of the manuscript.

  1. Please enhance the quality of Figure 1-4. Please provide references for elements not generated by the authors.

Response: We really appreciate the reviewer’s suggestion. As advised by the reviewer, we have updated the Figures 1-4 in the revised manuscript, such as enlarging the labeling and adjusting the composition of the figures. All the figures 1-4 in this manuscript were originally generated by us using BioRender.

  1. It is recommended that the authors to provide a paragraph of comments/reviews on the advantages/disadvantages of current engineering approach in ECM, EV, and GFs to control cellular activities and tissue regeneration.

Response: We thank the reviewer very much for this great suggestion. As advised by the reviewer, we have added one new paragraph in each section as 2.4, 3.4, 4.4 in the revised manuscript to review the advantages and disadvantages of current engineering approaches in ECM, EV, or GFs for tissue regeneration.

Reviewer 3 Report

This review describes novel strategies to utilize components of the microenvironment (ECM, EVs, GFs) of cells for supporting tissue regeneration. The review matches with the topic of the journal. It refers to many very different tissues and hence, sounds a bit less focussed on the first view. Looking at the mediators discussed it is closely confined to GFs, others like hormones, cytokines are not mentioned (line 45). Nevertheless, it summarizes some interesting examples of novel strategies and is clearly written.

4.2/4.3 about  GFs are rather short in comparison to the previous sections. The remodelling can lead to release of GFs and other bioactive compounds. Are there strategies developed to regulate the remodeling? (compare line 78-85)

Figures: they are helpful but the labeling is often too small.

Line 59: „fibrous forming proteins…proteoglycan“ the latter is an amorphe ECM component and does not form fibers.

Line 96: „decron“ does it mean decorin or a mimetic?

Line 139: „it was demonstrated the“ insert a „that“

Line 141: „activated biological signals“ which? „we has also“ write „have“

Line 142: insert a blank

Lines 189-191: the list should be ordered in a logical manner e.g. bring musculoskeletal disease and those affecting organs together.

Line 192: „short half-life“ – how short?

Line 214: „phosphorylation of Akt“ explain shortly the impact of phosphorylated Akt

Line 214-224: nearly every sentence used the verb „improved“ (style).

Line 233: „EC“ has this abbreviation been explained (endothelial cells?)

Line 253: „stability“ is a storage possible (freezing?) for how long?

Line 259: „native EVs“ is there a risk of immunogenicity?

Line 279: remove surplus blank

Line 337: „positively“ can it be substituted by „accelerated“?

Line 355: can „which“ be omitted?

Line 356: „clinical applications“ which strategies are indeed already used in the clinic? I think „that“ should be inserted (before „occurs“)?

Line 159: „real time biosensing“ might need some explanation.

Author Response

Reviewer 3:

This review describes novel strategies to utilize components of the microenvironment (ECM, EVs, GFs) of cells for supporting tissue regeneration. The review matches with the topic of the journal. It refers to many very different tissues and hence, sounds a bit less focused on the first view. Looking at the mediators discussed it is closely confined to GFs, others like hormones, cytokines are not mentioned (line 45). Nevertheless, it summarizes some interesting examples of novel strategies and is clearly written.

4.2/4.3 about  GFs are rather short in comparison to the previous sections. The remodeling can lead to release of GFs and other bioactive compounds. Are there strategies developed to regulate the remodeling? (compare line 78-85)

Response: We really appreciate the reviewer’s great comment. As the reviewer mentioned, ECM can interact directly with GFs, offering protection from degradation and controlling bioactivity of the growth factor (T Wilgus, Advances in Wound Care. 2012, 1(6): 249-254). ECM regulates the retention and presentation of GFs through electrostatic interactions. Electronegative components within the ECM (e.g., GAGs, HAp, etc.) have a high affinity for the positively charged amine groups of GFs (R Gresham et al., Bioactive Materials. 2021, 6: 1945–1956). Therefore, strategies have been designed to generate biomaterials with increased electronegative charge to mimic the high affinity of GFs to the ECM. Facca et al. demonstrated the presentation of BMP-2 and TGF-β1 from polyelectrolyte (PE) coated scaffold with a high negative electrostatic charge induced osteogenic differentiation of embryonic stem cells after subcutaneous implantation in mouse (S Facca et al., Proceedings of the National Academy of Sciences. 2010, 107(8): 3406-3411). Shah et al. showed that BMP-2 and VEGF loaded polyelectrolyte multilayer (PEM) films promoted bone regeneration upon subcutaneous implantation in rat (N Shah et al., Biomaterials. 2011, 32(26): 6183-6193). The information has been added at line 355-365 in 4.2 in the revised manuscript, and the related references also have been cited in the revised manuscript.

Figures: they are helpful but the labeling is often too small.

Response: We thank the reviewer very much for this helpful comment. As advised by the reviewer, we have updated all the figures and made the labels of the figures bigger and clearer in the revised manuscript.

Line 59: „fibrous forming proteins…proteoglycan“ the latter is an amorpha ECM component and does not form fibers.

Response: We appreciate the reviewer for this great comment. We have removed the “fibrous forming” from the line 59 in the revised manuscript.

Line 96: „decron“ does it mean decorin or a mimetic?

Response: We thank the reviewer for pointing out this typo. We meant decorin and have corrected it at line 96 in the revised manuscript.

Line 139: „it was demonstrated the“ insert a „that“

Response: As advised by the reviewer, we have inserted a “that” at line 141 in the revised manuscript.

Line 141: „activated biological signals“ which? „we has also“ write „have“

Response: We appreciate the reviewer for this comment. LLP2A modification can activate the integrin-mediated biological signals, such as focal adhesion kinase (FAK). We have updated this information at line 143 in the revised manuscript. We also corrected the “has” to “have” at line 143 in the revised manuscript.

Line 142: insert a blank

Response: A blank has been inserted at line 144 in the revised manuscript.

Lines 189-191: the list should be ordered in a logical manner e.g. bring musculoskeletal disease and those affecting organs together.

Response: As advised by the reviewer, we have reordered the list in a logical manner at line 203-205 in the revised manuscript.

Line 192: „short half-life“ – how short?

Response: We thank the reviewer for this question. EV has a short half-life of approximately few minutes. We have added this information at line 205-207 in the revised manuscript.

Line 214: „phosphorylation of Akt“ explain shortly the impact of phosphorylated Akt

Response: We appreciate this suggestion. Phosphorylation of Akt play key roles in multiple cellular processes such as cell apoptosis, cell proliferation, and cell migration. We have added this information at line 229-230 in the revised manuscript.

Line 214-224: nearly every sentence used the verb „improved“ (style).

Response: We thank the reviewer for this comment. As suggested by the reviewer, we have updated the related sentences and replaced “improve” with other various words at line 231-237 in the revised manuscript.

Line 233: „EC“ has this abbreviation been explained (endothelial cells?)

Response: We thank the reviewer for this nice comment. Yes, the full name of “EC” has been shown at line 109 in 2.2.

Line 253: „stability“ is a storage possible (freezing?) for how long?

Response: We thank the reviewer for this comment. Yes, as the reviewer mentioned, the storage stability is one challenge to translation and widespread application of EV-based therapeutics, which must be addressed to enable use of therapeutic EVs beyond resource-intensive settings. Studies to date suggest that the most promising mode of storage is −80°C, however, understanding of storage-mediated effects is still limited. Additionally, the effects of storage appear to vary with sample source.

Line 259: „native EVs“ is there a risk of immunogenicity?

Response: We thank the reviewer for this question. As nano-sized membrane structures naturally released by cells for intercellular communication, EVs are highly biocompatible with low cytotoxicity and immunogenicity, and therefore EVs are possibly to be a safe and effective therapeutic approach (V Xue et al., Expert Opinion on Biological Therapy. 2020, 7: 767-777). This information has been updated at line 274 in the revised manuscript.

Line 279: remove surplus blank

Response: Thank you. The surplus blank has been removed from line 294 in the revised manuscript.

Line 337: „positively“ can it be substituted by „accelerated“?

Response: We agree with the reviewer. As suggested by the reviewer, we have replaced “positively” with “accelerated” at line 379 in the revised manuscript.

Line 355: can „which“ be omitted?

Response: Thank you. We have deleted “which” at line 408 in the revised manuscript.

Line 356: „clinical applications“ which strategies are indeed already used in the clinic? I think „that“ should be inserted (before „occurs“)?

Response: We thank the reviewer for this comment. We have added several tissue-engineered products used in the clinic at line 407-408 in the revised manuscript. Also, we have inserted the “that” before “occurs” at line 409 in the revised manuscript.

Line 359: „real time biosensing“ might need some explanation.

Response: Thank you. The real time biosensor is an analytical device that combines a recognition element with a transducer to produce a measurable signal that can be correlated with the concentration of the analyte of interest, which could enable the analysis to be carried out at point-of-care (POC). This information has been added at line 412-413 in the revised manuscript.